# Acerola (*Malpighia* spp.) Waste: A Sustainable Approach to Nutraceutical, Pharmaceutical, and Energy Applications

**José Roberto Vega-Baudrit** [1,2,*], **Melissa Camacho** [1], **Diego Batista-Menezes** [1], **Yendry Corrales-Ureña** [1], **Juan Miguel Zúñiga** [1], **Arturo Mora Chacón** [3], **Nicole Lecot** [3], **Luis Castillo Henríquez** [1] and **Mary Lopretti** [3]

1 Laboratorio Nacional de Nanotecnología, Centro de Nacional de Alta Tecnología, LANOTEC-CeNAT-CONARE, Pavas, San José 10109, Costa Rica; mcamacho@cenat.ac.cr (M.C.); dbatista@cenat.ac.cr (D.B.-M.); ycorrales@cenat.ac.cr (Y.C.-U.); jzuniga@cenat.ac.cr (J.M.Z.); luis.castillo-henriquez@etu.u-paris.fr (L.C.H.)
2 Laboratorio de Polímeros POLIUNA, Universidad Nacional, Heredia 40101, Costa Rica
3 Laboratorio de Técnicas Nucleares Aplicadas a Bioquímica y Biotecnología, Centro de Investigaciones Nucleares, Facultad de Ciencias, Universidad de la República, Matojo 2055, Montevideo 11400, Uruguay; arturo.mora.chacon@est.una.ac.cr (A.M.C.); nlecot@fcien.edu.uy (N.L.); mlopretti@gmail.com (M.L.)
* Correspondence: jvegab@gmail.com

**Abstract:** This study reviews the many uses for waste generated from acerola (*Malpighia* spp.) production, a tropical fruit renowned for its nutrient-rich content. Traditionally considered an environmental burden, this waste is now gaining attention for its sustainable applications in green technology. This review outlines the extraction of valuable bioactive compounds, like polyphenols, carotenoids, and pectin, that can be extracted from the acerola fruit and acerola waste, and it also delves into its potential in materials science, particularly in the creation of pharmaceutical formulations, nanomaterials, composites, biofuels, and energy applications. On the medical front, the paper highlights the promise that acerola waste holds in anti-inflammatory, antihyperglycemic, and anticancer therapies. By outlining challenges and opportunities, the review emphasizes the untapped potential of acerola waste as a resource for high-value products. These findings suggest a paradigm shift, turning what has been considered waste into a sustainable asset, thereby encouraging environmentally responsible practices within the fruit industry.

**Keywords:** acerola; acerola waste; circular economy; nutraceutical; biopolymers



## 1. Introduction

Nowadays, there is an increase in the use of natural extracts from tropical fruits to meet the significant demand for natural ingredients instead of synthetic compounds [1]. Consequently, there is an increase in organic waste. Acerola extract is one of these products that has increased in production, and the global market for this extract is estimated to be US 1.5 million in the year 2023, and it is projected to reach US 40 million by 2033 [2]. China, India, and Brazil are the largest acerola producers, and the estimated production is higher than 40 million tons/year [3]. In recent decades, Brazil has established itself as the global leader in acerola production, with an estimated annual yield of 61,000 metric tons [4,5]. This dominant position has enabled Brazil to control a significant share of the international market for acerola-based products, which ranges from frozen fruit, juice, and jams to frozen concentrates and liqueurs. Moreover, the cultivation of acerola has been expanding to other parts of the Americas and even Europe, largely to produce ascorbic acid supplements and specialized fruit juices [5–8]. The large-scale production of acerola raises environmental concerns, especially concerning waste management. Byproducts, such as seeds, bagasse, ripe fruits, peels, and pulp, constitute approximately 40% of the fruit's total volume, meaning that 24,400 tons of waste could be produced in a country such as Brazil

and be utilized in high-valuable subproducts [9]. Residues from its processing also contain high amounts of vitamin C, phenolic, and other bioactive compounds [8].

The generation of acerola byproducts could contribute to environmental concerns if not managed efficiently. However, the high nutrient content in these byproducts indicates a missed opportunity for resource recovery. Preventing organic waste from becoming waste and creating high-value products will help us positively impact our environment.

Thus, ecofriendly strategies for recycling these byproducts into valuable products are under investigation. The multifaceted chemical composition of acerola byproducts has garnered interest in novel applications. Current research explores their utility in formulating nutraceuticals, skincare products, and natural dyes in the textile industry [10]. The diverse chemical constituents of acerola byproducts make them a promising resource for many applications across different industries.

Although these byproducts pose potential environmental risks if improperly managed, they also offer untapped opportunities for sustainable development. These residues can be harnessed to produce a range of valuable products, including natural preservatives and dietary supplements, as they are often rich in phenolic compounds and other bioactive substances, sometimes even more so than the edible portions of the fruit [10–12]. Emerging studies are starting to examine the potential of acerola in various health-related applications beyond its antioxidant capabilities. For instance, acerola extract has demonstrated anti-inflammatory and antimicrobial effects, suggesting a role in natural medicine and the potential for developing novel pharmaceuticals [13]. Further research is warranted to explore these bioactive properties in greater depth, including clinical trials that might confirm the translational value of these findings.

While acerola's high ascorbic acid content is well-documented, the fruit also serves as a rich source of other phytonutrients, including carotenoids, phenols, flavonoids, and anthocyanins [14,15]. These phytonutrients endow the fruit with various bio -functional and therapeutic properties that are yet to be comprehensively explored.

To fill this gap in our understanding and to potentially broaden the fruit's applications, this review aims to provide an in-depth analysis of the current state of acerola waste research. It will focus on its compositional attributes, pharmacological properties, and additional value-added uses, thereby helping to pinpoint current challenges and opportunities for its more extensive and economically viable industrial applications. Sustainable management strategies and innovative approaches could further elevate their value and utility. Figure 1 presents a scheme showing the type of industry and possible products that can be derived from the acerola, which will be discussed in this manuscript.

*Acerola Plant and Fruit*

The acerola (*Malpighia punicifolia*) (Figure 1) is an exotic tropical fruit that resembles the cherry. It is a member of the Malpighia genus within the Malpighiaceae family, comprising 45 species of shrubs or small trees. These plants are predominantly indigenous to the tropical and subtropical regions of the Americas [16,17]. The fruit is harvested from April to November, and the tree flowers are harvested after 3 to 4 weeks [7]. As early as 1946, researchers Asenjo and Guzmán identified acerola's exceptional ascorbic acid (vitamin C) content, thereby pioneering scientific interest in its potential health benefits [18]. Known for its rich nutrient profile, acerola contains antioxidant compounds, such as vitamins, carotenoids, and polyphenols [6,19], as well as thiamin, riboflavin, niacin, proteins, and mineral salts, mainly iron, calcium, and phosphorus [16]. This seminal work catalyzed subsequent studies and has cemented the fruit's reputation for its unique compositional attributes and functional significance.

Figures 2 and 3 show the anatomical description of the plant and the fruits, respectively. Furthermore, it shows the activity type of the compounds found in the plant part and tissues. A complete characterization of other parts and tissues of this plant could yield new and valuable information on other properties that could differ from those of other structures analyzed.

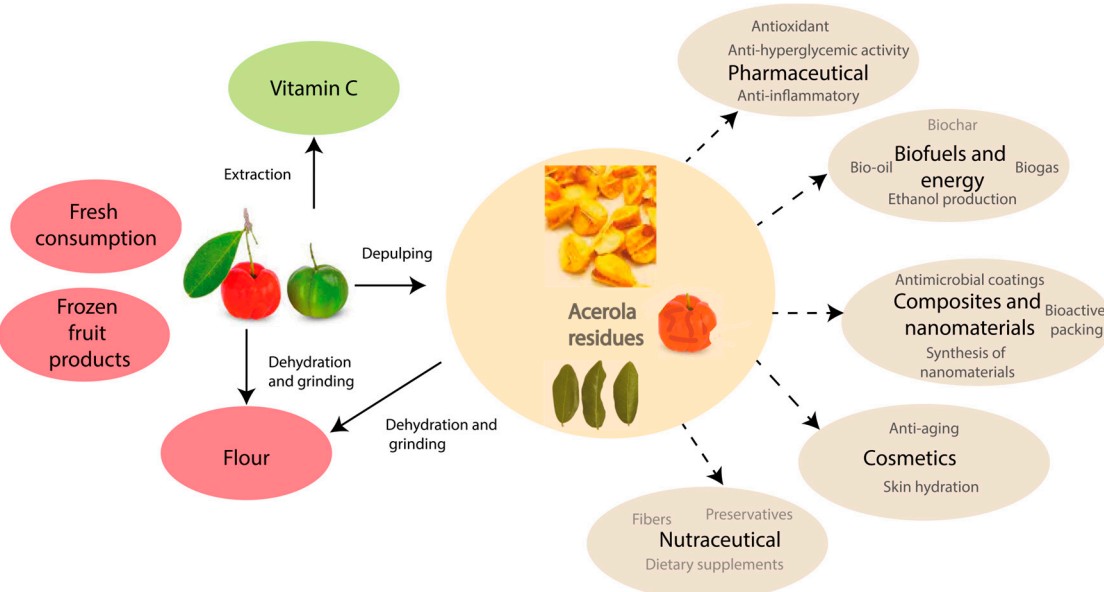

**Figure 1.** Acerola waste industry application opportunities.

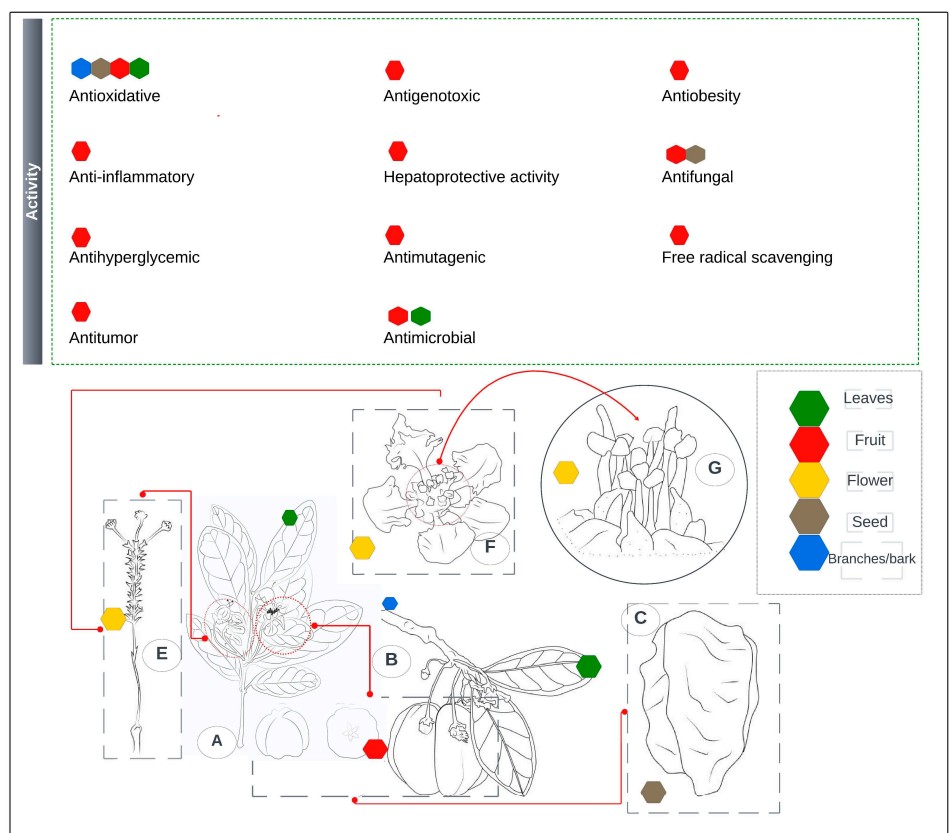

**Figure 2.** Anatomical structures in the genus *Malpighia*: (A) *M. emarginata* with fruiting branch and fruits; (B) approximation to fruiting branch for a most detailed view; (C) seed type—pyrene; (E) racemose inflorescence with bibracteolate peduncule from *M. romeroana*; (F) typical *Malpighia* flower; (G) approximation to *Malpighia* spp.'s reproductive structures (styles/stamens). Based on Woodson et al. (1980) and Brücher (1989) [20,21]. Colored hexagons identify the plant structure mentioned

above. The upper side shows the different activities associated with the acerola extracts. aNScientific reviews from the last decade are cited below, identified by the color according to the plant structure analyzed. No literature was found concerning the properties of the *Malpighia* spp. flowers (figure self-made). References for Activity: Antioxidative: [22–25]; Anti-inflamatory: [16], Antihyperglycemic: [16], antitumor: [26] Antigenotoxic: [27]; Hepatoprotective: [16,28,29], Antimutagenic: [16], Antimicrobial: [17,24], Antiobesity: [25]; Antifungal: [22,30], Free radical scavenging: [30].

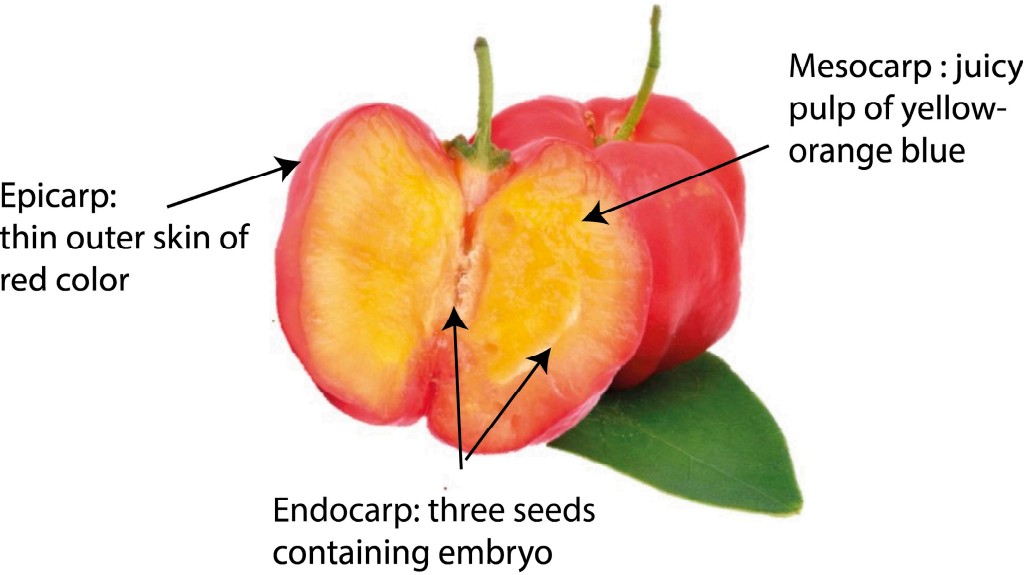

**Figure 3.** Morphological aspects of acerola fruit (*Malpighia emarginata*) and its associated structural elements (figure self-made).

## 2. Chemical Composition and Characterization of Acerola Wastes

Because of its composition, the acerola fruit exhibits a set of well-documented properties, like antioxidative, anti-inflammatory, antihyperglycemic, antitumor, antigenotoxic, hepatoprotective [16,31,32], antimutagenic, antiobesity [33], antimicrobial [34], antifungal, and free radical scavenging activity [24]. These bioactive properties have stimulated acerola consumption as fresh fruit or as processed products processed product [6]. Regarding these characteristics, acerola has great agro-industrial potential in markets that demand products rich in nutrients and, at the same time, improve health and prevent degenerative diseases [35,36], proving there is a large amount of research published on acerola in the food industry.

Furthermore, this fruit has been widely analyzed regarding its phytochemical and physicochemical compositions. Some authors, such as Prakash et al. (2018) [7], offer a comprehensive literature analysis focusing on the compositional characteristics of the fruit and the bifunctional properties of the present phytonutrients. Chang et al. (2018) and others studied the phytochemical compositions, antioxidant efficacies, and potential health benefits of acerola, among other fruits [37]. Belwal et al. (2018) reviewed the scientific research on bioactive chemical constituents and the beneficial health effects of acerola extracts and isolated compounds [16].

Poletto et al. (2021) reported the mass balance of the green acerola juice extraction process, obtaining 1.65% m/m of juice powder, 6.15% m/m of dry bagasse, and 1.93% dry nonpomace. They did not find ascorbic acid in the bagasse extract but a high amount in the non-pomace: 82% of the dry juice. Table 1 shows the amount of phytochemicals found in the bagasse and nonpomace reported by Poletto et al. (2021) [38] and Gualberto et al. (2021) [39].

**Table 1.** Phytochemical bioactive compounds found in acerola extract evaluated for antioxidant activity.

| Source | Bioactive Compound | Concentration |
|---|---|---|
| Bagasse | (9Z)-lutein | 3.40 ± 0 [38] a |
| | (All-E)-B-carotene | <0.05 ± 0 [38] a |
| | B-carotene | 5.84 ± 0.01 [39] b |
| | Carotenoids | 5.84 ± 0.01 [39] b |
| | Catechin | 22.46 ± 0.08 [39] b |
| | Epicatechin | 21.73 ± 0.35 [39] b |
| | Ferulic acid | 51.30 ± 3.73 [39] b |
| | Flavonoids total | 571.98 ± 40.29 [39] b |
| | Kaempferol | 27.59 ± 0.32 [39] b |
| | Naringenin | 810.40 ± 76.76 [39] b |
| | Organic acids | 1235.00 ± 25.00 [39] c |
| | P-coumaric acid | 59.48 ± 0.82 [39] b |
| | Rutin | 53.25 ± 2.81 [39] b |
| | Succinic acid | 119.00 ± 23.00 [39] c |
| | Tartaric acid | 1116.00 ± 2.00 [39] c |
| Juice powder | L-ascorbic acid | 314.50 ± 5.80 [38] a |
| Nonpomace | (9Z)-lutein | 11.10 ± 0 [38] a |
| | (All-E)-B-carotene | 5.80 ± 0 [38] a |
| | L-ascorbic acid | 158.00 ± 0.40 [38] a |

References = [ ]. Measurement units: a = mg/g, b = μg/g, and c = mg/100 g.

This elevated concentration of antioxidants, especially vitamin C, positions acerola as an economically significant commodity. It finds extensive applications in the food industry, primarily in producing natural preservatives and nutraceutical products. These attributes underscore the fruit's health benefits and enhance its market value and the potential for diverse applications of its bioactive compounds [10,35].

Acerola (*Malpighia punicifolia*) is a reservoir of multiple macro- and micronutrients, encompassing glucose, fructose, sucrose, and various organic acids [40,41]. The fruit's physicochemical attributes and nutritional valuation are contingent upon various factors, including geographical origin, environmental conditions, agronomic practices, maturation stage, and postharvest processing and storage [31,36]. The subsequent sections delineate the fruit's composition more exhaustively.

Carbohydrates: Byproducts of acerola are notably rich in carbohydrates, including glucose, fructose, sucrose, and starch as the predominant constituents [42,43]. These carbohydrates hold promise for utilization as energy sources or raw materials for producing biofuels and biopolymers [7].

Organic acids: Acerola fruit is renowned for its elevated concentrations of organic acids, a feature also found in its byproducts. Ascorbic acid (vitamin C) is the most abundant organic acid, followed by malic, citric, and tartaric acids [42,43]. These acids have prospective applications in the food and pharmaceutical sectors, serving as natural preservatives, flavor augmenters, and pH modulators [44]. The fruit ripeness influences the levels of ascorbic acid and sugar. Correa reported lower ascorbic acid and total sugar levels at full ripeness and higher sucrose levels [45]. Therefore, the volatile fraction of volatile compounds depends on the stages of fruit development [46].

Polyphenols: Acerola byproducts offer a potentially rich source of polyphenols known for their antioxidant and anti-inflammatory functionalities [47]. This category includes anthocyanins, flavonoids, and phenolic acids. Anthocyanins, water-soluble pigments

responsible for the red, blue, and purple hues in fruits, exhibit potential for application as natural colorants and therapeutic agents [48]. Flavonoids and phenolic acids also present various health benefits, such as protecting against cancer, cardiovascular diseases, and neurodegenerative disorders [48,49].

Carotenoids: Additionally, acerola byproducts are affluent in carotenoids, which confer various health benefits, including shielding against oxidative impairment and minimizing the risk of chronic diseases [50]. Beta-carotene predominates as the most abundant carotenoid, trailed by lutein and zeaxanthin. Carotenoids are promising for applications as natural colorants, food additives, and functional foods and dietary supplements [51].

Fatty acids: Acerola byproducts contain essential fatty acids like linoleic and oleic. These fatty acids are important in cellular function and inflammation modulation for human health. Moreover, these fatty acids could find applications in the cosmetics industry as moisturizing agents [52].

Pectins: The byproducts also encompass pectins, a class of soluble dietary fibers widely utilized in the food industry as thickening and gelling agents [53]. Pectins have been ascribed with cholesterol-lowering and prebiotic effects, making them pertinent for developing functional foods and dietary supplements [54,55].

Figure 4 shows some compounds that have been isolated from acerola.

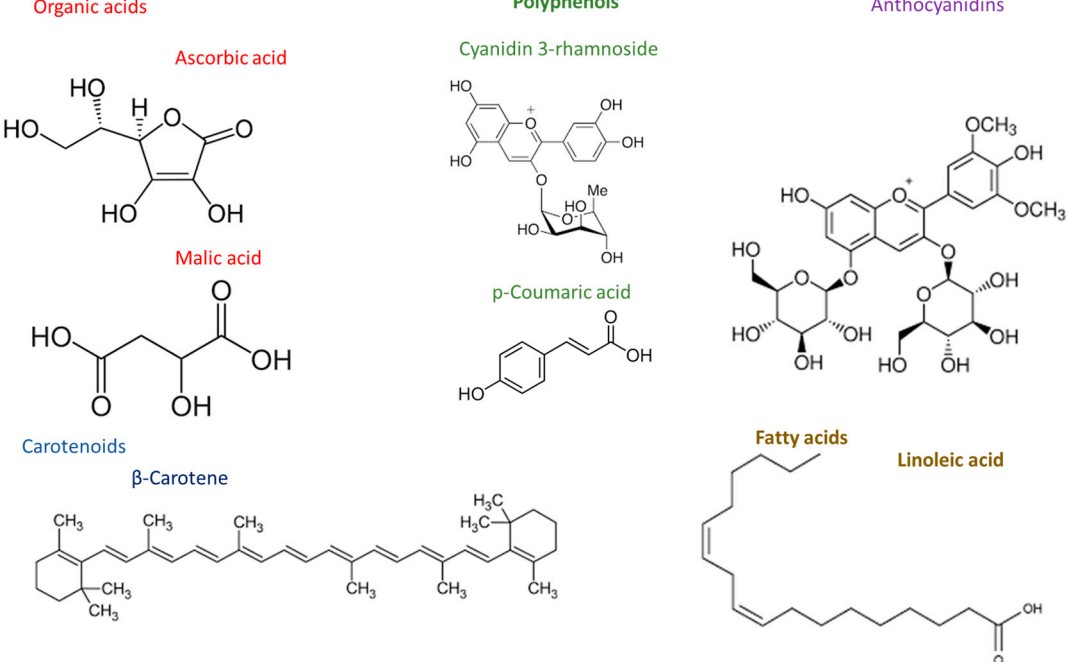

**Figure 4.** Chemical structure of some compounds reported to be present in the acerola: ascorbic acid [10], carotenoids [56], polyphenols [57,58], and fatty acids [52].

The chemical composition of acerola waste is not uniform and can exhibit considerable variability. This variability is influenced by several factors, including but not limited to the maturity stage of the fruit, the processing methodology employed, and the geographical location of cultivation. These factors can result in significant differences in the waste's macronutrients, micronutrients, and bioactive compounds.

Figure 5 shows a pie chart showing the % of the composition of acerola based on 100 g of fresh weight. Acerola is an important source of several macro- and micronutrients, such as ascorbic acid. It is one of the most important water-soluble vitamins, essential for collagen, carnitine, and neurotransmitter biosynthesis [7]. Table 2 shows pulped acerola residues' physical, chemical, and total antioxidant activity [59]. Volatile compounds such as furfural, hexadecanoic acid, 3-methyl-3-butenol, limonene, and 3-methyl-3-butenol have been isolated from acerola [60].

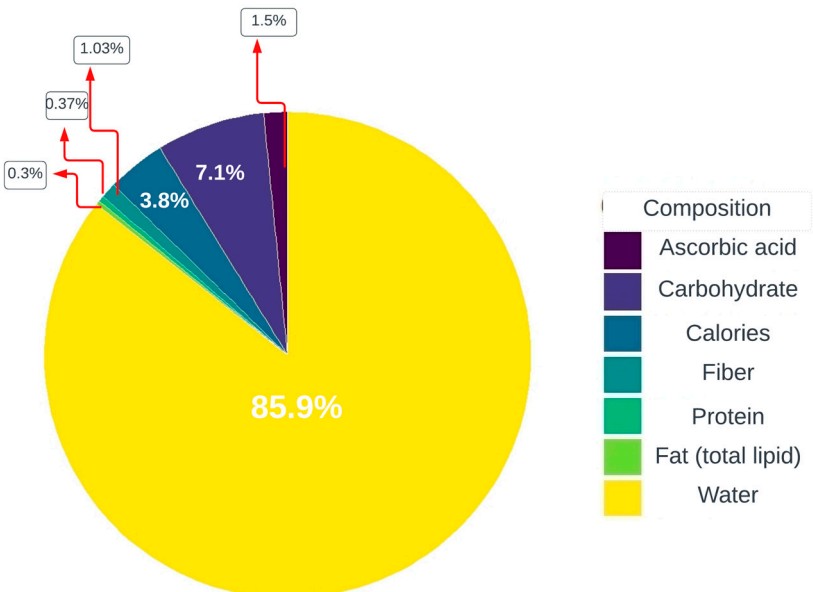

**Figure 5.** The composition of acerola (in 100 g fresh weight) [7].

**Table 2.** Pulped acerola residues' physical, chemical, and total antioxidant activity [59].

| Property or Activity | Value |
|---|---|
| pH | 3.38 |
| Total sugars (% glucose) | 1.93 |
| Total anthocyanins (mg/100 g) | 19.43 |
| Yellow flavonoids (mg/100 g) | 36.56 |
| Total polyphenols (mg gallic acid/100 g) | 545.98 |
| Total antioxidant activity (μM Trolox/g) | 17.70 |

## 3. Pharmaceutical and Nutraceutical Applications

In addition to the fruit, waste byproducts and other plant parts have been analyzed for their composition and properties. To mention some examples, Monteiro et al. (2020) studied the phytochemical and bromatological composition of acerola bagasse flour, which is rich in fiber and antioxidant compounds [61]. Da Silva et al. (2020) investigated bioactive compounds' content and extraction techniques from acerola waste, determining their potential in food and pharmaceutical applications [62]. Da Silva et al. (2020) studied the constituents in acerola leaves; this study proposes a phytochemical, nutritional, and immunostimulatory analysis of aqueous extracts [62]. Vasavilbazo et al. (2018) analyzed the leaves and barks of acerola to determine its phenolic composition, carotenoid content, and antioxidant activity [33].

However, many published articles focused on evaluating a specific biological activity, or if a more general investigation is developed, this topic needs to be addressed. This review aimed to detail all of the reported physical activities associated with acerola fruits, leaves, seeds, peel, and residual byproducts, highlighting its possible uses in the biomedical industry.

### 3.1. Antioxidant Activity

Antioxidant activity is one of the widest studied properties in acerola; it is proven that its phytochemicals and compounds represent a powerful source of natural antioxidants, which are health-promoting and disease-preventing activities [36]. The wide range of phytochemicals present in acerola contributes significantly to its high antioxidant capacity. This activity depends on the synergistic action of the constituents of these different fractions [14]. Because of the great interest in the antioxidant activity in acerola, the measurement of

its capacity was performed with DPPH●, ABTS+, FRAP, and ORAC assays, and multiple analyses were performed in different matrices; Table 3 shows some assays applied to other parts of acerola showing promising results on antioxidant capacity tests.

**Table 3.** Measurement of the antioxidant capacity with DPPH●, ABTS+, FRAP, and ORAC assays in different vegetal sources of acerola.

| Organic Sample | Extract Condition | DPPH | ABTS− | ORAC | FRAP | Source |
|---|---|---|---|---|---|---|
| **Leaves** | Methanolic | 10.88 + 0.38 [a] | 12.62 + 1.08 [a] | | | [33] |
| | Methanolic | 4.98 [a] | | | | [63] |
| | Saline | 38.59 + 1.20 [e] | | | 416.11 + 0.46 [e] | [22] |
| | Ethanolic | 1.75 + 1.06 [i] | 0.41 + 0.16 [i] | 3.875 + 0.18 [i] | | [64] |
| | Hydroethanolic | 68.8 + 0.18 [a] | | 38.1 + 0.03 | | [20] |
| | Methanolic | 21.69 + 1.62 [a] | 18.78 + 10.72 | | | [33] |
| | Methanolic | 1.625 + 0.9 [i] | 0.31 + 0.01 [i] | 1.63 + 0.9 [i] | | [64] |
| **Ripe fruit** | Water and methanol | 125.66 + 8.37 [b1] | 91.76 + 6.24 [b1] | 76.71 + 1.34 [b2] | | [14] |
| | Water and methanol | 82.22 + 2.9 | 64.57 + 2.47 | 57.77 + 1.61 | | [65] |
| | Water and methanol | 0.75 + 0.35 [i] | 0.305 + 0.02 [i] | 7.0 + 4.24 [i] | | [64] |
| **Unripe fruit** | Ethanolic | 1910 [c] | 8613.54 [c] | 2454.42 [c] | 1166.09 [c] | [23] |
| | Methanolic | 21.16 + 0.91 | 16.355 + 1.34 | | | [33] |
| | Ethanolic | 0.3 + 0.2 [i] | 0.28 + 0.01 [i] | | 5.0 + 0.0 [i] | [64] |
| | Methanolic | 0.375 + 0.18 [i] | 0.195 + 0.01 [i] | | 3.5 + 2.1 [i] | [64] |
| | Water | 0.375 + 0.18 [i] | 0.17 + 0.03 [i] | | | [64] |
| **Juice** | Seeds in water | | 0.18 + 0.013 [d] | | | [66] |
| | Pulp in water | | 3.44 + 0.12 [d] | | | |
| **Pulp** | Hydroethanolic | 7433 + 26.2 [e] | 8512 + 61.4 [e] | | | [27] |
| | Hydromethanolic | 19.65 + 1.92 [e] | 17.1 + 0.30 [e] | 11.815 + 0.37 | | [14] |
| | Hydroethanolic | | | | | |
| **Bark** | Methanolic | 15.36 + 0.18 [a] | 10.91 + 1.15 [a] | | | [33] |
| **Bagasse** | | | 16.14 + 0.01 [f] | | 0.92 + 0.01 [f] | [28] |
| | Hydromethanolic | | 405.11 + 1.83 | | | [67] |
| **Seeds/peels/barks mixed.** | Methanolic | | 790 + 14.00 [g] | | 2348.65 + 11.21 [g] | [68] |
| | Hydroacetone | 0.33 + 0.02 [h2] | 291.71 + 20.90 [h1] | | | [69] |
| | Hydroethanolic extract | 0.22 + 0.01 [h2] | 1445.10 + 73.07 [h1] | | | [69] |
| | Hydromethanolic extract | 0.23 + 0.01 [h2] | 1145.50 + 45.81 [h1] | | | [69] |
| | Hydroacetone | | 291.71 ± 20.90 [h1] | | | [69] |
| | Hydroethanolic | | 1445.10 ± 73.07 [h1] | | | [69] |
| | Hydromethanolic | | 1145.50 ± 45.81 [h1] | | | [69] |
| | Hydroacetone | 2228.51 + 6.29 [i] | 5180.85 + 184.785 [i] | | 45,325.64 + 2245.64 [i] | [39] |
| | Hydroethanolic extract | 2265.755 + 33.64 [i] | 1460.21 + 22.51 [i] | | 54,917.62 + 2007.535 [i] | [39] |
| | Hydromethanolic extract | 2305.83 + 37.355 [i] | 3349.055 + 90.69 [i] | | 41,023.735 + 3350.54 [i] | [39] |

[a] They calculate the antioxidant capacity as the amount of antioxidants needed to decrease by 50% the initial DPPH● concentration (EC50 (DPPH)), expressed as micrograms per milliliter ($\mu$g/mL) [33]. [b1] $TEAC_{ABTS}$ and $TEAC_{DPPH}$ values expressed as mM (mmol kg$^{-1}$ pulp, mmolL$^{-1}$ juice) [14]. [b2] TEAC values expressed as mM (mmol kg$^{-1}$ pulp, mmolL$^{-1}$ juice) [65]. [c] The analytical standard, Trolox, was used to construct the calibration curves, and all results are expressed as $\mu$M equivalent to Trolox/g of dry weight [23]. [d] The antioxidant potential of the sample was calculated from a standard curve of the synthetic antioxidant, Trolox, in the concentration range of 0 to 7 nmol; the results are expressed in mmol Trolox equivalents/100 mL sample [66]. [e] The Trolox calibration curve was constructed with concentrations varying from 0 to 250 $\mu$mol/L. The results are expressed in $\mu$mol of Trolox equivalent per 100 g of fresh weight ($\mu$mol Trolox/100 g fw) [27]. [f] The results are expressed as micromoles of Trolox equivalent antioxidant capacity (TEAC) per gram of sample ($\mu$mol TEAC/g) [28]. [g] The results are expressed as micromoles of Trolox equivalent antioxidant capacity (TEAC) per hundred grams of sample ($\mu$mol TEAC/100 g) [69]. [h1] The results are expressed as antioxidant activity equivalent to the Trolox ($\mu$Mol TEAC.g$^{-1}$ of the extract total phenolic) [69]. [h2] The results are expressed as extract phenolic ($\mu$Mol) DPPH.$\mu$Mol$^{-1}$ [69]. [i] The results are expressed in $\mu$mol of Trolox and mg of ascorbic acid per 100 g of dry residue [39].

### 3.2. Anti-Inflammatory Activity

Inflammation is a physiological response that triggers a defense mechanism against various stimuli or conditions. The inflammatory response is induced by a sequential release of inflammatory mediators and the recruitment of leukocytes to the inflammation site; once they arrive, they are activated, triggering the release of more mediators, creating a network of connections between the tissue and the cells of the immune system, thus resulting in the inflammatory response [70]. The phytochemicals present have been shown to reduce inflammation and related diseases through several points of cellular inflammatory pathways [71]. Some studies suggest that some metabolites in acerola are related to the modulation of pathways involved in regulating the inflammatory mechanism, such as mitogen-activated protein kinases (MAPKs) p38, ERK $\frac{1}{2}$, and JNK [72]. Different groups have investigated anti-inflammatory effects by performing in vitro assays using LPS-stimulated RAW-264.7 macrophage cell lines. Cabral et al. (2020) studied the anti-inflammatory properties of a blend powder formulation of 80% acerola and 20% green tea extract; their results suggest that cotreatment with blends could modulate the redox parameters in cells during the vitro inflammatory response [72]. Moreover, the cotreatment with blends modulated the inflammatory response by altering the secretion of cytokines like IL-1β, IL-6, IL-10, and TNF-α, causing, as a consequence, an anti-inflammatory effect. Albuquerque et al. (2019) studied four different fruit byproducts, including acerola (peel and seeds), to evaluate the anti-inflammatory properties of water extracts prepared with this biomass; the results showed that the aqueous extract of acerola does not present NO-reducing activity in the LPS-stimulated macrophages experiment [73]. However, the extract in the presence of fruit fibers promotes the adhesion of beneficial probiotics to the intestinal walls, maintaining a balanced oxidative state in the intestinal environment, thus protecting the epithelium against inflammatory processes. Pereira et al. (2020) studied the anti-inflammatory activity using freeze-dried acerola extracts. Their results show that by evaluating selected inflammation-associated mRNA markers, a 50 ug/mL acerola extract can inhibit the expression of IL-6, IL-1β, and COX-2 genes associated with the in vitro inflammatory process [74]. A different approach was developed by Milanez et al. (2014); these authors carried out an in vivo study using male Swiss albino mice and food treatments supplemented with commercial acerola juice. After 13 weeks, the histological analysis showed that acerola juice restores metabolic and inflammatory pathways to an average level [75].

### 3.3. Antihyperglycemic Activity

Hyperglycemia is a condition, like type 2 diabetes, characterized by an abnormal postprandial increase in blood glucose level caused by regularly consuming rapidly digestible carbohydrates [76]. An important therapeutic approach to the treatment of type 2 diabetes is to decrease postprandial hyperglycemia by slowing glucose absorption through inhibition of the enzymes α-amylase and α-glucosidase in the digestive tract [76]. Polyphenols have shown some beneficial effects in the prevention and management of diabetes through the inhibition of glucose absorption, inhibition of digestive enzymes, regulation of intestinal microbiota, modification of the inflammation response, and inhibition of the formation of advanced glycation end products [77]. Barbalho et al. (2011) studied the effects of acerola juice intake on the glycemic and lipid profile of diabetic and nondiabetic Wistar rat offspring [78]. The biochemical profile determination of blood samples taken from acerola-extract-treated rats indicates a significant reduction in glucose, cholesterol, and triglycerides and increased HDL-c, promising a potential strategy for preventing diabetes and other diseases. Hanamura et al. (2005) performed research to characterize polyphenols from acerola fruits, identifying as a result three different polyphenols (C3R, P3R, and quercitrin) for the first time [57]. This study proved that three isolated, purified polyphenols and a crude extract of mixed polyphenols have an inhibitory effect on the α-amylase enzyme [57]. Hanamura et al. (2006) studied the antihyperglycemic effect of a natural ethanol polyphenol fraction, evaluating the impact on the glucose intake of Caco-2 cells. They demonstrate that this extract decreased the glucose uptake level dose-

dependently by adding the acerola polyphenol fraction. In this study, the researchers also performed an assay on ICR male mice, analyzing glucose and maltose intake; the results showed that the crude acerola extracts significantly suppressed the plasma glucose level after administering both glucose and maltose, suggesting that this extract had a preventive effect on hyperglycemia in the postprandial state [79].

### 3.4. Antitumor Activity

Epidemiological research suggests that a polyphenol-rich diet protects against tumors, inhibiting proliferation, inducing apoptosis, and reducing drug resistance in gastric cancer cells [80]. The role of phytochemicals has been associated with the modulation of different signaling cascades of the cell division process, which, in turn, is related to the induction of apoptosis, suppression of epithelial to mesenchymal transition, and subsequent metastatic behavior of cancer cells. In addition, studies suggest that different phytochemicals increase the sensitivity of cancer stem cells to chemotherapy drugs [81]. Motohashi et al. (2004) studied organic solvent extractions of acerola using two human oral tumor cell lines (oral squamous cell carcinoma cells (HSC-2) and submandibular gland carcinoma cells (HSG) and normal oral cells as control; their results suggest that some of the hexane fractions inhibited Pgp function in multidrug-resistant cancer cells more effectively than the control verapamil, highlighting the tumor-specific cytotoxic potential of acerola extract [82]. Nagamine et al. (2002) studied the capacity of an acerola extract to control cell proliferation and the activation of the Ras signal pathway in the promotion stage of lung tumorigenesis in mice. They injected the mice with 4-(methyl nitrosamine)-1-(3-pyridyl)-1-butanone (NNK), a potent carcinogen, and then fed the mice with acerola extract; the results showed that the pretreatment with acerola inhibited the increase in the levels of proliferating nuclear cell antigen and ornithine decarboxylase at the promotion stage and regulates abnormal cell growth at the promotion stage of lung tumorigenesis in mice [83]. Ribeiro et al. (2015) demonstrated that some anthocyanins, such as cyanidin 3-glucoside and cyanidin 3-rutinoside, lutein, $\alpha$-carotene, and $\beta$-carotene, present in acai fruit extracts attenuate chemically induced mouse colon carcinogenesis by increasing total GSH and attenuating DNA damage and preneoplastic lesion development; this is noteworthy, as these compounds have been widely identified in acerola [84].

### 3.5. Antigenotoxic Activity

Metal ions generate DNA damage directly or indirectly by producing reactive oxygen species; these compounds are closely related to some diseases, since some free radicals in an oxidative stress condition are not neutralized by antioxidant compounds or antioxidant cell protective mechanisms [85]. In their research, Nunes et al. (2016) demonstrated that comparing bone marrow samples from mice groups treated with $FeSO_4$ against groups, in turn, pretreated with acerola juice, it is proven that the juice exerted antimutagenic activity by significantly decreasing the mean values of micronuclei in the bone marrow [85]. Da Silva et al. (2011) studied the antigenotoxic effect of acerola pulp extracts from ripe and unripe fruits. They applied a comet essay in CF-1 male mice blood to establish the protective ability of fruit extracts against the oxidative stress induced by hydrogen peroxide; the results suggest that unripe fruit presented higher DNA protection than ripe fruit extract due to a high content of vitamin C in unripe fruits [86]. In a similar study, Dimer et al. (2013) analyzed the antigenotoxic capacity of acerola juice from fruits at different stages of ripening [87]. Subjecting male mice to a diet high in sugars and fat induced glucose intolerance and DNA damage in the specimens. After 13 weeks, the diet was supplemented with acerola juice, resulting in a partial reversal of the DNA damage in the blood, kidney, liver, and bone marrow caused by the diet. Antigenotoxic properties of acerola may be affected by genetic diversity as well as environmental factors [25]. Differences in antigenotoxic activity exist among acerola varieties grown in different regions of Brazil. Extracts of acerola pulp were evaluated in a comet assay on mouse blood cells in vitro, demonstrating some significant differences in DNA protection against $H_2O_2$ damage

between two different plantations [68], as well as among fruits of the same acerola species, same ripening stage, and harvest time, from two different plantations [25].

### 3.6. Hepatoprotective Activity

Reactive nitrogen and oxygen species are produced during metabolic reactions and exert many important functions in cellular defense mechanisms; however, these components are overpowered in pathological situations, which can generate adverse effects [88]. As mentioned, antioxidants play a protective role by inhibiting free radical-induced reactions and reducing oxidative damage, preventing the peroxidative deterioration of the cell membrane lipids and DNA damage [27]. Nagamine et al. (2004) studied the hepatoprotective effect of dry extract powders of fruit purees and grind leaves of acerola [89]. Researchers applied different extract powders in a physiological salt solution after the intoxication induced by D-Galactosamine in male Wistar rats. As a result, the components of the extracts diminished the hepatic inflammatory response, decreased hepatocellular injury, and improved liver function in rats under GalN intoxication [89]. El-Hawary et al. (2021) evaluated a dried ethanol powdered acerola leaf extract on rats induced with hepatic damage by CCL4. Their results showed that all the tested doses showed a higher reduction in serum levels of TNF-$\alpha$ [24]. However, the dose of 800 mg/Kg showed the highest hepatoprotective effect as it reduced the elevated serum levels of ALT, AST, NO, and TNF-$\alpha$ liver content, increasing the serum level of catalase. Marques et al. (2018) studied the protective potential of a lyophilized extract of acerola bagasse against CCL4-induced hepatoxicity in Wistar rats; their results showed that the treatments with this extract presented a decrease in the activity of aspartate aminotransferase, alanine aminotransferase, and gamma-glutamyl transferase, and an increase in superoxide dismutase, total antioxidant capacity, and albumin content [90]. Gomes et al. (2013) investigated the effects of ripe acerola juice in vivo using female Swiss mice. The animals were pretreated with acerola juice for 15 consecutive days and then subjected to ethanol-induced stress, evaluating serum enzymes and the degree of lipid peroxidation. Compared to the ethanol-only treated group, the liver of acerola juice-fed animals before acute ethanol administration showed a significant reduction of lipid peroxidation, similar to control levels, proving that acerola juice can prevent hepatic damage [27].

### 3.7. Antimutagenic Effect

Antimutagenic compounds reduce the frequency of spontaneous or induced mutations; vitamin C, polyphenols, and carotenoids can reduce oxidative stress levels and quench free radicals, reducing the damage and promoting the protection of DNA against oxidative damage, showing antioxidant and antimutagenic effects [86,87]. Some compounds, such as cyclophosphamide (CP) [91], iodine-131 [92], and hydrogen peroxide [93], may result in undesirable side effects and cause cellular mutations in organisms. Acerola juice mixed with other fruits showed high antiproliferative and antimutagenic activities against alterations induced by cyclophosphamide, possibly attributed to its high content of bioactive compounds [94]; acerola pulp juice showed potential as an antimutagenic by statistically reduced the percentages of chromosomal alterations induced by this compound [91]. Research developed by Düsman et al. (2016) also demonstrated that the consumption of acerola showed antimutagenic effects against iodine-131 in acute and subchronic treatments, mainly by acting in the capture of free radicals produced by radiation [91]. Research developed by Almeida et al. (2014) also demonstrates the potential of acerola extracts to reduce 131-I-induced damage [92]. Some in vitro performed in bone marrow cells of mice stated that acerola mixed with different fruits show antimutagenic activity at different concentrations [94]; juices from ripe and unripe acerola showed a significant decrease in micronucleus in contrast with an animal group treated with FeSO$_4$ [85]. Using acerola juice as a food supplement can help decrease oxidative stress and protect or repair the damage to the DNA in obese animals [87]. Spada et al. (2008) studied the antimutagenic effect of different tropical fruits in *Saccharomyces cerevisiae*; their results suggest that frozen fruit

extracts exhibited antimutagenic effects and CAT-like activity. CAT neutralizes hydrogen peroxide, avoiding hydroxyl radical formation and damage to the DNA [93].

### 3.8. Antibacterial Activity

Secondary metabolites produced by plants have a defensive function when certain microorganisms, such as viruses, parasites, fungi, or bacteria, attack them. The compounds with antibacterial action are usually terpenoids, phenolic compounds, alkaloids, polypeptides, coumarins, and camphor [34,36,95,96]. The antimicrobial activity of different acerola extracts has been evaluated using methods such as agar diffusion assay, disc diffusion test, and minimal inhibitory concentration evaluation (MIC) assay. Motohashi et al. (2004) used different solvents to prepare extracts from frozen and fresh acerola fruits, showing relatively high activity against *S. epidermidis*, *E. coli*, and *P. aeruginosa* [82]. Montero et al. (2020) studied the antimicrobial properties of hexane extracts from pulp, barks, and seeds. Their results showed that pulp and seed extracts have the greatest activity against *E. coli* compared to all the other fruits tested. However, this inhibition was lower than the ampicillin control [95]. The bark extracts showed a low inhibition against *S. Typhimurium* and *C. albicans*. Some research groups proposed alcoholic extractions of phenolic compounds from different parts of the plant and industrial byproducts to prove their potential against common bacteria. An ethanol extract of acerola leaves showed the ability to mildly inhibit the growth of *B. subtilis*, *S. aureus*, *E. coli*, and *P. aeruginosa* when compared to an ampicillin control [24], methanol extractions of acerola bagasse flour exhibited antimicrobial activity on *L. monocytogenes*, *E. coli*, *P. aeruginosa*, and *S. cholerasuis* [97]; in addition, methanolic fractions of different phenolic compounds were evaluated, the results suggest that the flavonoid fraction showed moderate antimicrobial properties against *S. aureus* [34,36]. Some research has focused on evaluating the potential of hydroalcoholic extractions. A 30:70 $v/v$ ratio of ethanol:water extract yields strong antimicrobial activity against *B. thermophacta*, *P. fluorescent*, and *P. fragi* and moderate activity against *P. Putida* [98], and 1:1 solutions of ethanol:water showed moderate activity over *P. aeruginosa* and *L. monocytogenes* [99]. A different approach to evaluate antibacterial properties was proposed by Pinheiro et al. (2020) in which the development of nanoparticles from acerola waste products was shown to possess antimicrobial properties against *E. coli* [100].

### 3.9. Antiobesity

Obesity is excessive fat accumulation in body tissues, influenced by genetics, energy-dense intake, high-fat foods, and the absence of physical activity [67,101]. Excessive body fat gain promotes inflammatory conditions associated with producing pro-inflammatory cytokines, increased production of reactive oxygen species, and decreased antioxidant defenses [67]. Studies have shown the beneficial effects of antioxidant supplementation in the diet of obesity-related diseases [102] and the consumption of phytochemicals, which can help to reduce body weight, decrease white adipose tissue, and regulate the appetite hormone [75]. Milanez et al. (2014) studied the effect of unripe and ripe acerola juice on male Swiss mice fed with a cafeteria diet [75]. Their results showed a reduction in weight gain, a reduction in (TAG) levels, and an increase in important anti-inflammatory cytokines, such as IL-10 and TNF-$\alpha$, in adipose tissue. These results suggest that acerola juice reduces inflammation and diminishes obesity-associated defects in lipolytic processes. Vital et al. (2021) investigated the effects of acerola-derived products in the diet of male Wistar rats [101]. Treatment with acerola did not alter weight gain, energy efficiency, and body weight. In addition, it was shown to increase antioxidant activity and reduce adiposity, showing a novel alternative to treating obesity. The role of antioxidant activity is reinforced by research such as that of Dimer et al. (2017), showing that acerola juice can reduce oxidative stress caused by obesity on energy metabolism enzymes [102]. Marques et al. (2016) evaluated acerola bagasse flour methanol extracts to investigate the effects of -amylase and -glycosidase enzymes. Because of the content of phenolic compounds, the methanol extract inhibited these enzymes on in vitro assays; these results suggest that this

extract may represent a good source of inhibitors and can be used as an auxiliary in the treatment of obesity [67].

### 3.10. Antifungal Activity

Antifungal activity has been evaluated using the same techniques applied in antimicrobial assays. Motohashi et al. (2004) proved butanol and methanol extracts of acerola against yeast *C. albicans* and *C. glabrata*; their results showed a high inhibitory zone in contrast with X and X as positive controls [82]. Different studies evaluate the effects of acerola leaf extracts against different fungal species. Schmourlo et al. (2007) studied an aqueous extract, proving that this extract has antibacterial effects over T. rubrum through MIC and agar diffusion assay [96]. Moreover, Barros et al. (2019) tested aqueous saline extracts from dry leaves on different species of *Candida*. The saline extract proved effective by inhibiting 90% of *C. albicans*, *C. parapsilosis*, *C. krusei*, and *C. tropicalis* but ineffective against *C. glabrata* [22]. El-Hawary et al. (2022) studied the potential of ethanol extracts, showing a low response over *C. albicans* in contrast with the inhibition zone caused by the antibiotic amphotericin B [24].

Figure 6 shows the activity associated with each pharmaceutical application.

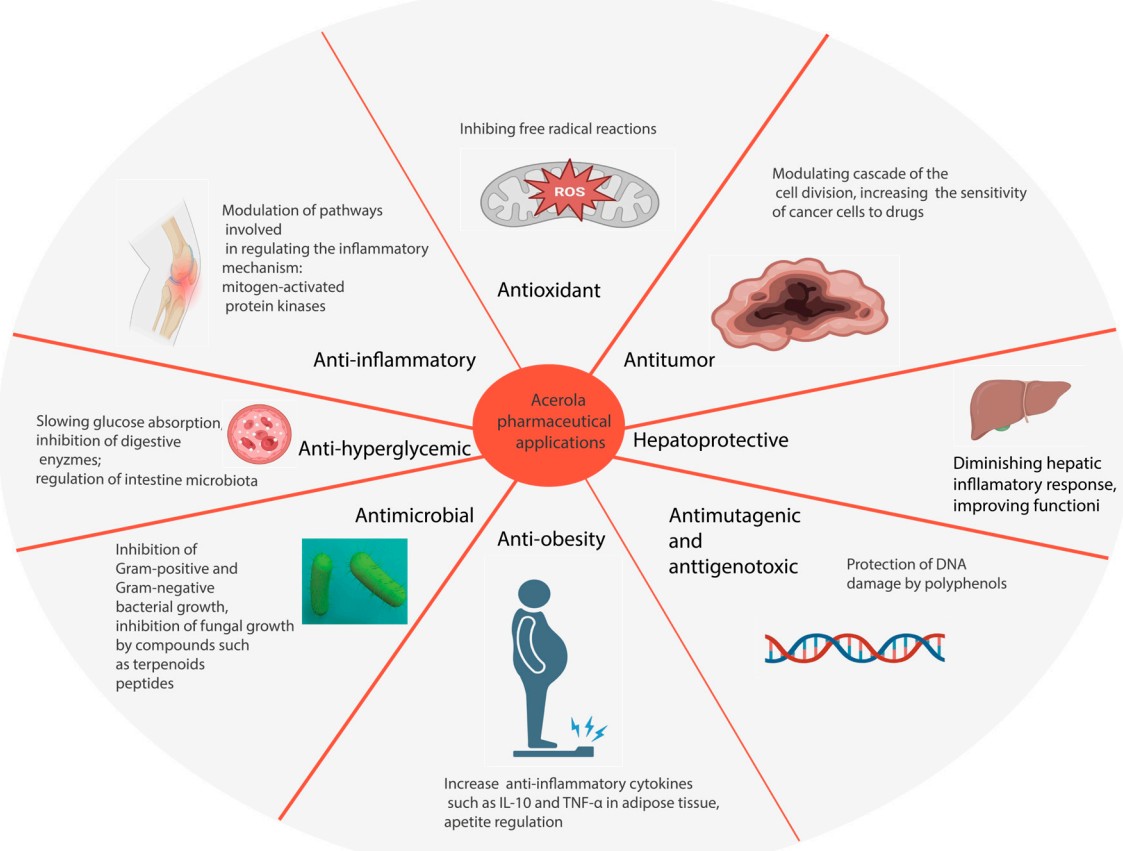

**Figure 6.** Pharmaceutical application of acerola (created with BioRender.com by the authors).

## 4. Nutraceutical

Acerola fruits are rich in bioactive compounds with various health benefits, making them a potential source for functional foods, but further research is needed for evidence-based product development [16].

Among the antioxidant compounds, Sousa (2010) highlighted that the acerola pulp residues had the highest amounts of carotenoids (881.56 µg/100 g), flavonoids (8.84 µg/100 g), and total phenolics (247.62 mg gallic acid/100 g fruit pulp residue) [103]. Further-

more, Magalhães et al. (2021) determined a protein percentage of 9.12% for acerola flour, which was similar to that found by Borges (2011), which was 9.45% [104]. Similarly, Soares et al. (2001) determined a value of 9.12% for the flour made with acerola residue and 9.05% for the powdered fruit pulp, indicating a similarity in the protein content between the fruit and the residue [105]. In this way, it is possible to consider that acerola flours can be used for human consumption, since they have a mix of bioactive compounds, proteins, vitamins, and fibers in their composition. Monteiro et al. (2020) focused on the potential use of waste from acerola juice production, which is abundant in fiber, carotenoids, and anthocyanins. The aim was to prepare and analyze flour made from these byproducts for nutritional and phytochemical qualities, intending to introduce it as a new ingredient in the food sector. The waste was sourced from The Federal Institute of Rio Grande do Norte, dried, and ground to create the flour. Nutritional and phytochemical assessments indicated that the flour is fiber-rich (77%) and has notable antioxidant properties due to its high levels of carotenoids and anthocyanins. The findings suggest that this acerola byproduct flour could serve as a nutritional supplement or food ingredient [61]. The byproducts generated from the commercial handling of acerola (*Malpighia emarginata* D.C.) are often inadequately managed or underutilized despite their rich concentration of antioxidant substances [106].

The pulp residues contain a high total dietary fiber content (>70%). Consequently, it can be an additive for different food products [106]. Abud et al. studied the addition of 10% residue incorporation in cookies to increase the nutritional value of food for poor populations, showing good acceptance by the consumer [106]. Abud also reported poly-galacturonase production by semisolid state fermentation at a small scale and concluded that the residue is feasible for producing this enzyme. However, further large-scale production studies are necessary to demonstrate the feasibility.

## 5. Cosmetics

Additionally, the fruit has been recognized for its potential in cosmetic applications. With its rich phytonutrient profile, acerola is currently being studied for its anti-aging effects on the skin. Preliminary investigations indicate that its bioactive compounds may protect against UV radiation damage and improve skin hydration, opening new avenues for the cosmetics industry to explore [107]. Focusing on product development in this sector could add another layer to the fruit's economic significance and further diversify its applications. The effects on melanogenesis of the acerola polyphenols were studied. The skin-lightening effect on brownish guinea pigs subjected to controlled UVB irradiation was studied; resulting in lightened skin pigmentation and reduced melanin content in B16 melanoma cells. Furthermore, the polyphenols inhibit mushroom tyrosinase activity [108]. Skin whitening and cell renewal treatments produced with the acerola cherry fermentation have been patented [109,110].

## 6. Biofuels and Energy Applications

Nogueira et al. (2019) explored the utility of Hydrothermal Carbonization (HTC) for optimizing the treatment and application of waste from acerola processing. Various factors, including temperature, time duration, biomass-to-water ratio, pH levels, and mixing speed, were evaluated for their impact on the efficiency of the HTC process and the properties of the resulting hydrochar. Optimal conditions were determined through a multicriteria evaluation method. The hydrochar exhibited a higher concentration of oxygen-bearing functional groups than similar biomass types cited in existing literature. Further experiments assessed the hydrochar's effectiveness in absorbing methylene blue dye, revealing peak absorption and removal efficiency conditions. The data suggest that the hydrochar's absorption is heat-driven and spontaneous, indicating that acerola waste could be transformed via HTC into a valuable resource for absorbing environmental pollutants [111].

De Oliveira reported obtention of ethanol from the hydrolysis of baggase from acerola and further fermentation of the glucose obtained. They pretreated them with diluted

sulfuric acid and enzymatic hydrolysis with *Trichoderma reesei*, obtaining glucose ranging from 0.78–1.95 g/L. Saccharomyces cerevisiae was used for fermentation and ethanol production [112].

The anaerobic digestion of the waste can be a source of biogas and methane. Barros et al. reported the anaerobic codigestion of acerola (*Malphigia emarginata*) agro-industry effluent with domestic sewage at mesophilic and thermophilic conditions have been reported, achieving biodegradability above 60% at mesophilic conditions. CODS removals and their conversion into methane above 50% and 45% in mesophilic conditions, respectively [113].

Two methods to produce important energy gas sources, such as biological hydrogen by photo-fermentation, where the dominant biocatalyst is photosynthetic bacteria, and dark-fermentation, fermentative anaerobic bacteria produce hydrogen in the absence of light, have not been yet explored [114].

## 7. Composites and Nanomaterials

Leon (2014) focused on synthesizing and characterizing a nanocomposite material comprising titanium dioxide and botanic and fruit extracts in nanoparticulate form (within a size range of 1–100 nm). This advanced material could be synthesized using an impregnation technique, wherein organic functional groups, inorganic moieties, and phytoextracts—such as could be the case with acerola extracts—could be adsorbed onto the titanium dioxide scaffold [115]. The result is an agent with pronounced antimicrobial properties, including disinfectant and antiseptic effects, that are efficacious against a broad spectrum of pathogens: bacteria, fungi, mycobacteria, spores, protozoa, and viruses. The product is a liquid suspension containing the solid nanomaterial in question.

The impregnation methodology is employed to achieve homogeneous dispersion of the functional groups and extract nanoparticles, while intramolecular interaction stability is maintained by temperature control during the synthesis process [115].

The antimicrobial activity profile of this conjugated nanomaterial is intrinsically linked to several parameters, including the particle size of the oxide scaffold, chemical functionalization, and surface dispersion of adsorbed extracts. Possible functional groups for incorporation include hydroxyl, carboxyl, amine, sulfate, and phosphate moieties. Furthermore, although the study focuses on titanium dioxide as the scaffold material, the technology could be extendable to other metallic oxides, such as silica, zirconia, zinc oxide, and alumina, without being exclusively restricted to these functional groups, extracts, or scaffolds. Therefore, this nanomaterial has applicability across diverse fields, from medicine to green technologies, offering an innovative and sustainable approach to pathogen management [116]. Reinaldo et al. reported the addition of grape skins and acerola residues to the cassava thermoplastic starch, developed a bioactive packing with improving antioxidant characteristics [116]

## 8. Waste Processing

M. Silva et al. (2020) focused on utilizing ultrasound-assisted methods to extract these antioxidants from leftover acerola material. Various factors, like the concentration of ethanol in the hydroethanolic mix, duration of extraction, operational temperature, and the ratio of liquid to solid, were assessed for their impact on total phenolic and flavonoid content and antioxidant capabilities. Optimal conditions for extraction were determined using a multicriteria optimization approach known as the desirability function. High-performance liquid chromatography with ultraviolet detection (HPLC-UV) was employed to pinpoint the primary antioxidant elements extracted under these conditions. The findings indicate that acerola byproducts offer promising avenues for enhanced utilization, particularly in the food and pharmaceutical sectors [42].

Nóbrega et al. (2015) investigated the impact of hot air drying on acerola waste's properties and nutritional content. The research explores various drying conditions, such as 60, 70, and 80 °C temperature settings and air speeds of 4, 5, and 6 m/s, comparing them to untreated acerola waste [117]. The study quantifies the retention of key bioactive elements

like phenolic compounds, carotenoids, anthocyanins, proanthocyanidins, and ascorbic acid after drying. The antioxidant properties were also measured. The findings indicate that the dried acerola waste retains significant levels of these important compounds, making it a viable source of health-promoting ingredients. This research offers a scientific foundation for the better use of acerola waste, especially given the growing issue of waste generation from agricultural production, notably in Brazil—one of the world's largest agricultural hubs. The dried acerola byproduct thus holds the potential for sustainable, health-focused applications in the food industry [117].

For another instance, there are studies that have examined the impact of different agricultural practices on soil organic carbon (C) and nitrogen (N) levels in Chapada da Ibiapaba, Brazil [118]. They compare organic acerola fruit farming and pastureland with conventional crop rotation systems featuring carrot, beet, and corn. Soil samples were taken from various depths and analyzed for indicators, including total organic C and N, microbial C and N, and mineralizable C [118]. The findings show significant improvements in organic and pasture systems compared to conventional and native forest soil, particularly in microbial C and N stocks. For instance, the organic acerola system had a 585% increase in Nmic stock compared to native forest soil, while conventional farming saw reductions of up to 59% in Cmic stocks. These metrics suggest that organic and pasture management practices sustain and effectively improve soil quality and carbon sequestration. Therefore, these sustainable approaches should be considered for better soil health in the Chapada da Ibiapaba region [118].

## 9. Conclusions

Acerola (*Malpighia* spp.) is an economically significant tropical fruit, primarily due to its rich content of antioxidants, particularly ascorbic acid. This study emphasizes the fruit's diverse applicability, extending from the food industry to energy and biofuels, with emerging possibilities in natural medicine and cosmetics. Although Brazil is at the forefront of global acerola production, large-scale cultivation yields considerable waste, accounting for approximately 40% of the fruit's total volume.

Rather than merely posing an environmental challenge, these residues stand as an untapped resource for sustainable development. Our findings indicate that these byproducts are abundant in bioactive compounds such as phenolics and other phytonutrients. Moreover, burgeoning research is unveiling the health benefits of acerola beyond its antioxidant capacity, including anti-inflammatory and antimicrobial properties. We did not identify processing as a significant challenge because fruit and vegetable companies have created processes to reutilize this type of waste. Acerola waste is nontoxic and almost 100% reusable as a raw material for new products, as has been demonstrated in flour production; the primary challenge is to separate the bioactive compounds of interest without degrading them and/or to find synergies with other stabilizing compounds to create long lasting products. Improving the management of acerola waste due to its biodegradability and water content is a key factor that can catalyze the creation of new products and markets for these residues. Also, the consumer sensibilization to buying products from sources considered waste without losing efficacy.

It is critical to highlight that these residues could serve as valuable assets within green technology, with potential applications in bioactive compound extraction, biopolymer development, and biofuel generation. Therefore, the effective utilization of acerola waste addresses environmental challenges and introduces economic value to what has traditionally been considered waste.

Sustainable management systems focused on using acerola and its byproducts could play a significant role in environmental conservation and soil quality improvement and contribute to carbon sequestration in specific regions. This study underscores the necessity for additional research to bridge the gaps in our understanding of acerola's multifaceted applications and its potential in sustainable waste management.

The present investigation is a valuable resource for future research, providing scientific justification for the more effective exploitation of this tropical fruit, aligning agricultural production with sustainability goals and added economic value.

**Author Contributions:** Conceptualization, J.R.V.-B., Y.C.-U. and D.B.-M.; investigation, J.R.V.-B., Y.C.-U., D.B.-M., J.M.Z., M.L., M.C., A.M.C., N.L. and L.C.H., writing—original draft preparation, J.R.V.-B., Y.C.-U., J.M.Z., D.B.-M., M.L., M.C., A.M.C., N.L. and L.C.H., writing—review and editing, J.R.V.-B., Y.C.-U., D.B.-M. and J.M.Z.; supervision, J.R.V.-B. All authors have read and agreed to the published version of the manuscript.

**Funding:** This research received no external funding.

**Institutional Review Board Statement:** Not applicable.

**Informed Consent Statement:** Not applicable.

**Data Availability Statement:** The data presented in this study are available in article.

**Conflicts of Interest:** The authors declare no conflict of interest.

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
