# Peer review of "Acerola (Malpighia spp.) Waste: A Sustainable Approach to Nutraceutical, Pharmaceutical, and Energy Applications"

_recycling, doi:10.3390/recycling8060096_

Round 1

Reviewer 1 Report

Comments and Suggestions for Authors

The submission description is “Biopolymer Development from Acerola Waste" While this work overall fits the scope of this journal, I would recommend a thorough revision before it can be considered for publication.

  1. This manuscript needs to have a central "story" or "message”, and everything included needs to contribute towards that. I think it needs some more work to develop a strong "story". 
  2. The introduction needs to be a little bit more elaborative about the need for green technology, especially wound healing, which is mentioned in the abstract. (doi.org/10.1016/j.mtsust.2022.100272, doi.org/10.1016/j.mtchem.2022.100910, doi.org/10.1016/j.ijbiomac.2022.10.169)
  3. This manuscript lacks good figures. The number of figures is not enough.
  4. Tables 1 and 2 can be presented in the form of a chart in a more beautiful way.
  5. The table is not adequate; in the table, the listed papers are short, not enough for a good review.
  6. The conclusion section should be improved by adding information regarding restrictions and limitations.

Comments on the Quality of English Language

Minor editing of English language required.

Author Response

General comments:

The references were increased from 85 to 120. All highlighted in yellow was changed with respect to the first version.

Reviewer 1

The submission description is “Biopolymer Development from Acerola Waste" While this work overall fits the scope of this journal, I would recommend a thorough revision before it can be considered for publication.

    1. This manuscript needs to have a central "story" or "message”, and everything included needs to contribute towards that. I think it needs some more work to develop a strong "story". 
    2. The introduction needs to be a little bit more elaborative about the need for green technology, especially wound healing, which is mentioned in the abstract. (doi.org/10.1016/j.mtsust.2022.100272 , doi.org/10.1016/j.mtchem.2022.100910 , doi.org/10.1016/j.ijbiomac.2022.10.169 )
    3. This manuscript lacks good figures. The number of figures is not enough.
    4. Tables 1 and 2 can be presented in the form of a chart in a more beautiful way.
    5. The table is not adequate; in the table, the listed papers are short, not enough for a good review.
  • The conclusion section should be improved by adding information regarding restrictions and limitations.

Thank you for taking the time to review our manuscript entitled "Biopolymer Development from Acerola Waste". We appreciate your constructive feedback and have made significant efforts to improve the manuscript according to your suggestions.

  1. Central "Story" or "Message": We agree that the manuscript needs a more focused central message. We have revised the text to centralize around the development and potential applications of products derived from acerola waste, emphasizing its importance in green technology and sustainable waste management. We add a Part 5 called Other applications **No medical applications**. We re/called part 4 as Pharmaceutical and nutraceutical applications

  1. Introduction and Green Technology: We have expanded the introduction to more elaborately discuss the need for green technology (part 5). We re-write the abstract to include some details:

Line 18: The study reviews the many uses for waste generated from acerola (Malpighia spp.) production, a tropical fruit renowned for its nutrient-rich content. Traditionally considered an environmental burden, this waste is now gaining attention for its sustainable applications in green technology.  This review outlines the extraction of valuable bioactive compounds like polyphenols, carotenoids, and pectin that can be extracted from the acerola fruit and acerola waste, and it also delves into its potential in material science, particularly in the creation of pharmaceutical formulations, nanomaterials, composites, biofuels, and energy applications. On the medical front, the paper highlights the promise that acerola waste holds as anti-inflammatory, anti-hyperglycemic activity, and anticancer therapies. By outlining challenges and opportunities, the review emphasizes the untapped potential of acerola waste as a resource for high-value products. These findings suggest a paradigm shift, turning what has been considered waste into a sustainable asset, thereby encouraging environmentally responsible practices within the fruit industry.

  1. Figures:
    In accordance with your comments, we have added  five new  figures to enrich the manuscript visually. These figures are:

Figure 1.  Acerola waste industry application opportunities

Figure. 2. Anatomical structures in the genus Malpighia. A) M. emarginata with fruiting branch and fruits, B) Approximation to fruiting branch for a most detailed view, C) Seed type pyrene,  E) Racemose inflorescence with bibracteolate peduncule from M.romeroana, F) Typical Malpighia flower, G) Approximation to Malpighia spp reproductive structures (styles/stamens). Based  in Woodson et al. (1980) and Brücher (1989) [24, 25]. Colored hexagons identify the plant structure mentioned above. The upper side shows the different activity associated with acerola extracts. Scientific Reviews from the last decade are cited below, identified by color according to the plant structure analyzed. No literature was found about the properties of Malpighia spp flowers. (figure self-made). 

Figure 4. Chemical structure of some compounds reported to be present in the acerola: ascorbic acid [39, 40], carotenoids [54], polyphenols [55 ][56], and organic acids [40].

Figure 5. The composition of acerola ( in 100 g fresh weight) [6].

Figure 6. Pharmaceutical application of acerola (Created with BioRender.com by the authors)..

  1. Tables 1 was changed   for figure 5. Now table 1 was substituted by:

Poletto et al. 2021 reported the mass balance of the green acerola juice extraction process, obtaining 1.65 %m/m of juice powder. 6.15 % m/m of dry bagasse and  1.93% dry non-pomace. They did not find ascorbic acid in the bagasse extract but a high amount in the non-pomace. Table 1 shows the amount of phytochemicals found in the bagasse and non-pomace reported by Poletto et al. 2021 [35] and Gualberto et al. 2021 [36].

                             Table 1. Phytochemical bioactive  compounds found in acerola extract evaluated for        antioxidant activity.             

Source

Bioactive compound

Concentration

Bagasse

(9Z)-lutein

158,00±0,40 [35]

(all-E)-β-carotene

314,50±5,80 [35]

β-carotene

169,00±5,80 [35]

Carotenoids

277,70±4,60 [35]

Catechin

0,03±0,00 [35]

Epicatechin

0,04±0,00 [35]

Ferulic acid

11,1 [35]

Kaempferol

5,8 [35]

L-ascorbic acid

<0,05 [35]

L-ascorbic acid

119,00±23,00 [36]

L-ascorbic acid

5,84±0,01 [36]

Naringenin

51,30±3,73 [36]

Organic acids

59,48±0,82 [36]

P-coumaric acid

22,46±0,08 [36]

Rutin

810,40±76,76 [36]

Non pomace

Succinic acid

53,25±2,81 [36]

Tartaric acid

571,98±40,29 [36]

*Measurement units: [35]) mg/100 g extract, [36]) µg/g extract..

Furthermore, more references were added to table 3, line 281.

  1. Adequacy of the Table: Not changed it.
  1. Conclusion Section:

We have revised the conclusion to discuss the restrictions and limitations of our study, offering a more balanced perspective of our findings.

Acerola (Malpighia spp.) is an economically significant tropical fruit, primarily due to its rich content of antioxidants, particularly ascorbic acid. This study emphasizes the fruit's diverse applicability, extending from the food industry to energy and biofuels, with emerging possibilities in natural medicine and cosmetics. Although Brazil is at the forefront of global acerola production, large-scale cultivation yields considerable waste, accounting for approximately 40% of the fruit's total volume.

Rather than merely posing an environmental challenge, these residues stand as an untapped resource for sustainable development. Our findings indicate that these by-products are abundant in bioactive compounds such as phenolics and other phytonutrients. Moreover, burgeoning research is unveiling the health benefits of acerola beyond its antioxidant capacity, including anti-inflammatory and antimicrobial properties. We did not identify the processing as a significant challenge because fruit and vegetable companies have created processes to reutilize this type of waste. The acerola waste is non-toxic and almost 100 reusable as a raw material for new products, as has been demonstrated by flour production; the primary challenge is to separate the bioactive compounds of interest without degrading them and/or to find synergies with other stabilizing compounds to create long lasting products. Improving the management of acerola waste due to its biodegradability and water content is a key factor that can catalyze the creation of new products and markets for these residues. Also, the consumer sensibilization to buying products from sources considered waste without losing efficacy.

It is critical to highlight that these residues could serve as valuable assets within green technology, with potential applications in bioactive compound extraction, biopolymer development, and biofuel generation. Therefore, the effective utilization of acerola waste addresses environmental challenges and introduces economic value to what has traditionally been considered waste.

Sustainable management systems focused on using acerola and its by-products could play a significant role in environmental conservation and soil quality improvement and contribute to carbon sequestration in specific regions. This study underscores the necessity for additional research to bridge the gaps in our understanding of acerola's multifaceted applications and its potential in sustainable waste management.

The present investigation is a valuable resource for future research, providing scientific justification for the more effective exploitation of this tropical fruit, aligning agricultural production with sustainability goals and added economic value.

Reviewer 2 Report

Comments and Suggestions for Authors

The manuscript is entitled " Biopolymer Development from Acerola (Malpighia spp.) Waste: A Sustainable Approach to Nutraceutical, Pharmaceutical Applications and Environmental Preservation”. The objective of this research was to provide a comprehensive overview of recent studies concerning the potential applications of acerola residues. These applications encompass the creation of nanomaterials, chitosan, and biofuels, in addition to the extraction of bioactive compounds. Furthermore, the potential medicinal applications of acerola wastes, including their anticancer, wound-healing, and antioxidant properties, are examined in the paper. The final section of this study consists of an analysis of the prospects and difficulties connected with the utilization of acerola waste as a sustainable source of both energy and raw materials. The manuscript is well done and planned. But some points should be addressed.

In the paper, the acerola data on environmental preservation features, as well as the use of nanomaterials, chitosan, and biofuels, has to be completed in depth.

It is important to provide specific information on the plant's biopolymers in detail.

Not only should the names of the active phytochemical substances such as aceronidin, leucocyanidin, norfriedelin be mentioned but the portion that was employed should also be specified. The information can be summarized in the form of a table.

Italicization is required for many of the scientific names of species included in the MS.

The application must to cover a greater extent than only the biological components.

Comments on the Quality of English Language

 Editing of  language required

Author Response

  1. In the paper, the acerola data on environmental preservation features, as well as the use of nanomaterials, chitosan, and biofuels, has to be completed in depth.

We added specific section, reporting the literature in the field suggested by the reviewer. 

Line 565: 4. Nutraceutical

Acerola fruits are rich in bioactive compounds with various health benefits, making them a potential source for functional foods, but further research is needed for evidence-based product development [16].

Among the antioxidant compounds, Sousa (2010) highlighted that the acerola pulp residues had the highest amounts of carotenoids (881.56 μg/100 g), flavonoids (8.84 μg/100 g), and total phenolics (247.62 mg gallic acid/100 g fruit pulp residue) [104]. Furthermore, Magalhães et al. (2021) determined a protein percentage of 9.12% for acerola flour, which was similar to that found by Borges (2011), which was 9.45% [103]. Similarly, Soares et al. (2001) determined a value of 9.12% for the flour made with acerola residue and 9.05% for the powdered fruit pulp, indicating a similarity in the protein content between the fruit and the residue [104]. In this way, it is possible to consider that acerola flours can be used for human consumption since they have a mix of bioactive compounds, proteins, vitamins, and fibers in their composition. Monteiro et al. (2020) focused on the potential use of waste from acerola juice production, which is abundant in fiber, carotenoids, and anthocyanins. The aim was to prepare and analyze flour made from these byproducts for nutritional and phytochemical qualities, intending to introduce it as a new ingredient in the food sector. The waste was sourced from The Federal Institute of Rio Grande do Norte, dried, and ground to create the flour. Nutritional and phytochemical assessments indicated that the flour is fiber-rich (77%) and has notable antioxidant properties due to its high levels of carotenoids and anthocyanins. The findings suggest that this acerola byproduct flour could serve as a nutritional supplement or food ingredient [105]. The byproducts generated from the commercial handling of acerola (Malpighia emarginata D.C.) are often inadequately managed or underutilized despite their rich concentration of antioxidant substances [106]

The pulp residues contain a high total dietary fiber content (>70%). Consequently, it can be an additive for different food products [57]. Abud et al. studied the addition of  10 % residue incorporation in cookies to increase the nutritional value of food for poor populations, showing good acceptance by the consumer [107]. Abud also reported polygalacturonase production by semi‐solid state fermentation at a small scale and concluded that the residue is feasible for producing this enzyme. However, further large-scale production studies are necessary to demonstrate the feasibility.

5.Cosmetics

Additionally, the fruit has been recognized for its potential in cosmetic applications. With its rich phytonutrient profile, acerola is currently being studied for its anti-aging effects on the skin. Preliminary investigations indicate that its bioactive compounds may protect against UV radiation damage and improve skin hydration, opening new avenues for the cosmetics industry to explore [108]. Focusing on product development in this sector could add another layer to the fruit's economic significance and further diversify its applications. The effects on melanogenesis of the acerola polyphenols were studied. The skin-lightening effect on brownish guinea pigs subjected to controlled UVB irradiation was studied; resulting in lightened skin pigmentation and reduced melanin content in B16 melanoma cells. Furthermore, the polyphenols inhibit mushroom tyrosinase activity [109]. Skin whitening and cell renewal treatments produced with the acerola cherry fermentation have been patented [110, 111]

  1. Biofuels and energy applications

 Nogueira et al. (2019) explored the utility of Hydrothermal Carbonization (HTC) for optimizing the treatment and application of waste from acerola processing. Various factors, including temperature, time duration, biomass-to-water ratio, pH levels, and mixing speed, were evaluated for their impact on the efficiency of the HTC process and the properties of the resulting hydrochar. Optimal conditions were determined through a multi-criteria evaluation method. The hydrochar exhibited a higher concentration of oxygen-bearing functional groups than similar biomass types cited in existing literature. Further experiments assessed the hydrochar's effectiveness in absorbing methylene blue dye, revealing peak absorption and removal efficiency conditions. The data suggests that the hydrochar's absorption is heat-driven and spontaneous, indicating that acerola waste could be transformed via HTC into a valuable resource for absorbing environmental pollutants [112].

 De Oliveira reported obtention of ethanol from the hydrolysis of baggase from acerola and further fermentation of the glucose obtained. They pre-treated them with diluted sulfuric acid and enzymatic hydrolysis with Trichoderma reesei, obtaining glucose ranging from 0.78–1.95 g/L. Saccharomyces cerevisiae was used for fermentation and ethanol production  [113].

The anaerobic digestion of the waste can be a source of biogas and methane. Barros et al. reported the anaerobic co-digestion of acerola (Malphigia emarginata)  agro-industry effluent with domestic sewage at mesophilic and thermophilic conditions have been reported, achieving biodegradability above 60% at mesophilic conditions. CODS removals and their conversion into methane above 50% and 45% in mesophilic conditions, respectively [114].  Two methods to produce important energy gas sources, such as biological hydrogen by photo-fermentation, where the dominant biocatalyst is photosynthetic bacteria, and dark-fermentation, fermentative anaerobic bacteria produce hydrogen in the absence of light, have not been yet explored [115].

  1. . Composites and nanomaterials

Leon, 2014 focused on synthesizing and characterizing a nanocomposite material comprising titanium dioxide and botanic and fruit extracts in nanoparticulate form (within a size range of 1-100 nm). This advanced material could be synthesized using an impregnation technique, wherein organic functional groups, inorganic moieties, and phytoextracts—such as could be the case with acerola extracts— could be adsorbed onto the titanium dioxide scaffold [116]. The result is an agent with pronounced antimicrobial properties, including disinfectant and antiseptic effects, that are efficacious against a broad spectrum of pathogens: bacteria, fungi, mycobacteria, spores, protozoa, and viruses. The product is a liquid suspension containing the solid nanomaterial in question.

The impregnation methodology is employed to achieve homogeneous dispersion of the functional groups and extract nanoparticles, while intramolecular interaction stability is maintained by temperature control during the synthesis process [116].

The antimicrobial activity profile of this conjugated nanomaterial is intrinsically linked to several parameters, including the particle size of the oxide scaffold, chemical functionalization, and surface dispersion of adsorbed extracts. Possible functional groups for incorporation include hydroxyl, carboxyl, amine, sulfate, and phosphate moieties. Furthermore, although the study focuses on titanium dioxide as the scaffold material, the technology could be extendable to other metallic oxides, such as silica, zirconia, zinc oxide, and alumina, without being exclusively restricted to these functional groups, extracts, or scaffolds. Therefore, this nanomaterial has applicability across diverse fields, from medicine to green technologies, offering an innovative and sustainable approach to pathogen management [116]. Reinaldo et al. reported the addition  of grape skins and acerola residues to the cassava thermoplastic starch, developed a bioactive packing  with  improving  antioxidant characteristics [117]

  1. Waste processing

  1. Silva et al. (2020) focused on utilizing ultrasound-assisted methods to extract these antioxidants from leftover acerola material. Various factors, like the concentration of ethanol in the hydroethanolic mix, duration of extraction, operational temperature, and the ratio of liquid to solid, were assessed for their impact on total phenolic and flavonoid content and antioxidant capabilities. Optimal conditions for extraction were determined using a multi-criteria optimization approach known as the desirability function. High-Performance Liquid Chromatography with Ultraviolet detection (HPLC-UV) was employed to pinpoint the primary antioxidant elements extracted under these conditions. The findings indicate that acerola byproducts offer promising avenues for enhanced utilization, particularly in the food and pharmaceutical sectors [118].

Nóbrega et al. (2015) investigated the impact of hot air drying on acerola waste's properties and nutritional content. The research explores various drying conditions, such as 60, 70, and 80°C temperature settings and air speeds of 4, 5, and 6 m/s, comparing them to untreated acerola waste [119]. The study quantifies the retention of key bioactive elements like phenolic compounds, carotenoids, anthocyanins, proanthocyanidins, and ascorbic acid after drying. The antioxidant properties were also measured. The findings indicate that the dried acerola waste retains significant levels of these important compounds, making it a viable source of health-promoting ingredients. This research offers a scientific foundation for the better use of acerola waste, especially given the growing issue of waste generation from agricultural production, notably in Brazil—one of the world's largest agricultural hubs. The dried acerola byproduct thus holds the potential for sustainable, health-focused applications in the food industry [119].

For another instance, there are studies to examine the impact of different agricultural practices on soil organic carbon (C) and nitrogen (N) levels in Chapada da Ibiapaba, Brazil [121]. It compares organic acerola fruit farming and pastureland with conventional crop rotation systems featuring carrot, beet, and corn. Soil samples were taken from various depths and analyzed for indicators, including total organic C and N, microbial C and N, and mineralizable C [121]. The findings show significant improvements in organic and pasture systems compared to conventional and native forest soil, particularly in microbial C and N stocks. For instance, the organic acerola system had a 585% increase in Nmic stock compared to native forest soil, while conventional farming saw reductions of up to 59% in Cmic stocks. These metrics suggest that organic and pasture management practices sustain and improve soil quality and carbon sequestration more effectively. Therefore, these sustainable approaches should be considered for better soil health in the Chapada da Ibiapaba region [120].

  1. It is important to provide specific information on the plant's biopolymers in detail.

Figure 2 shows specific information and details about the activity of the compounds extract from each plant part.

Figure 2. Anatomical structures in the genus Malpighia. A) M. emarginata with fruiting branch and fruits, B) Approximation to fruiting branch for a most detailed view, C) Seed type pyrene,  E) Racemose inflorescence with bibracteolate peduncule from M.romeroana, F) Typical Malpighia flower, G) Approximation to Malpighia spp reproductive structures (styles/stamens). Based  in Woodson et al. (1980) and Brücher (1989) [24, 25]. Colored hexagons identify the plant structure mentioned above. The upper side shows the different activity associated with acerola extracts. Scientific Reviews from the last decade are cited below, identified by color according to the plant structure analyzed. No literature was found about the properties of Malpighia spp flowers. (figure self-made). 

  1. Not only should the names of the active phytochemical substances such as aceronidin, leucocyanidin, norfriedelin be mentioned but the portion that was employed should also be specified. The information can be summarized in the form of a table.

Table 1 was changed for thisn new table , and this information was added.

Poletto et al. 2021 reported the mass balance of the green acerola juice extraction process, obtaining 1.65 %m/m of juice powder. 6.15 % m/m of dry bagasse and  1.93% dry non-pomace. They did not find ascorbic acid in the bagasse extract but a high amount in the non-pomace. Table 1 shows the amount of phytochemicals found in the bagasse and non-pomace reported by Poletto et al. 2021 [35] and Gualberto et al. 2021 [36].

                             Table 1. Phytochemical bioactive  compounds found in acerola extract evaluated for        antioxidant activity.             

Source

Bioactive compound

Concentration

Bagasse

(9Z)-lutein

158,00±0,40 [35]

(all-E)-β-carotene

314,50±5,80 [35]

β-carotene

169,00±5,80 [35]

Carotenoids

277,70±4,60 [35]

Catechin

0,03±0,00 [35]

Epicatechin

0,04±0,00 [35]

Ferulic acid

11,1 [35]

Kaempferol

5,8 [35]

L-ascorbic acid

<0,05 [35]

L-ascorbic acid

119,00±23,00 [36]

L-ascorbic acid

5,84±0,01 [36]

Naringenin

51,30±3,73 [36]

Organic acids

59,48±0,82 [36]

P-coumaric acid

22,46±0,08 [36]

Rutin

810,40±76,76 [36]

Non pomace

Succinic acid

53,25±2,81 [36]

Tartaric acid

571,98±40,29 [36]

*Measurement units: [35]) mg/100 g extract, [36]) µg/g extract..

  1. Italicization is required for many of the scientific names of species included in the MS.

It was checked.

  1. The application must cover a greater extent than only the biological components.

New sections were added to the manuscript to complement this part.

Reviewer 3 Report

Comments and Suggestions for Authors

The paper „ Biopolymer Development from Acerola (Malpighia spp.) Waste: A Sustainable Approach to Nutraceutical, Pharmaceutical Applications and Environmental Preservation” regards an issue of potential uses of acerola wastes, including seeds, grains, and pulp. This review was based on 85 publications, most of which were related to the current research. In parts 2 and 3, the authors characterised the composition of acerola wastes in terms of chemical composition and macro and micronutrients.  In the fourth part, the authors presented several bioactive properties of compounds extracted from acerola waste, such as antioxidants, anti-inflammatory agents, antitumor agents, and many others. This part highlights the potential of using bio-active compounds extracted from different acerola waste but still doesn’t cover the main topic of the presented publication – biopolymer development. There is no clear evidence that acerola might be a source of any biopolymer or biocomposite. Additionally, there is no specific comment on the challenges and opportunities in utilising acerola waste as a sustainable source of raw materials and energy, as pointed out in the abstract, introduction and conclusion part.

The article moderately covers the use of acerola waste. Still, it lacks a critical commentary on the prospects for using this research, the profitability of the process and the challenges associated with introducing such solutions to the market, which should be included in the review article. Before publication, the issues mentioned above should be completed by the authors.

Author Response

Reviewer 3:

The article moderately covers the use of acerola waste. Still, it lacks a critical commentary on the prospects for using this research, the profitability of the process and the challenges associated with introducing such solutions to the market, which should be included in the review article. Before publication, the issues mentioned above should be completed by the authors.

More economic aspects were added to the text:

Line 33: Nowadays, there is an increase in the use of natural extracts from tropical fruits to meet the significant demand for natural ingredients instead of synthetic compounds. Consequently, there is an increase in organic waste. The acerola extract is one of these products that has increased its production, and the global market extract was estimated to be US$1.,5 Million in the year 2023, and it is projected to reach US$40 Million by 2033 [1]. China, India, and Brazil are the largest acerola producers, and the production estimation is higher than 40 million tons/year [1].  In recent decades, Brazil has established itself as the global leader in acerola production, with an estimated annual yield of 61,000 metric tons [3, 4]. This dominant position has enabled Brazil to control a significant share of the international market for acerola-based products, which range from frozen fruit and juice to jams, frozen concentrates, and liqueurs. Moreover, the cultivation of acerola has been expanding to other parts of the Americas and even Europe, largely to produce ascorbic acid supplements and specialized fruit juices [5-7]. The large-scale production of acerola raises environmental concerns, especially concerning waste management. By-products such as seeds, bagasse, ripe fruits, peels, and pulp constitute approximately 40% of the fruit's total volume, meaning that 24400 tons of waste could be produced in a country such as Brazil and be utilized in high-valuable sub-products [13].  Residues from its processing also contain high amounts of vitamin C, phenolic, and other bioactive compounds.

Also in line 166 was reported the mass balance:

Poletto et al. 2021 reported the mass balance of the green acerola juice extraction process, obtaining 1.65 %m/m of juice powder. 6.15 % m/m of dry bagasse and 1.93% dry non-pomace. They did not find ascorbic acid in the bagasse extract but a high amount in the non-pomace, 82% of the dry juice. Table 1 shows the amount of phytochemicals found in the bagasse and non-pomace reported by Poletto et al. 2021 [35] and Gualberto et al. 2021 [36].

Reviewer 4 Report

Comments and Suggestions for Authors

The review paper by J. R. Vega-Baudrit describes an utility of the tropical fruit acerola including all its constituents like pulp, outer skin, and seeds as source of vitamins, important nutrients (super-food) as well as a platform for diverse drug preparation towards a wide range of dangerous diseases. This manuscript is rather tutorial and a bit popular-scientific one encompassing some botanical and medicine aspects, nevertheless interesting for a broad readership. Despite the manuscript title, there is very scarce information and consideration related to Environmental Preservation in the Manuscript. So, it should be supplemented by analysis of the modern literature related to acerola wastes recycling, e.g., https://doi.org/10.1016/j.wasman.2020.03.037. Different ways of this recycling should be considered, for instance, thermochemical treatment affording biochar, and concentrate more on chemical and technological aspects of recycling.

The authors should also to point out the position of this review among relevant literature in introduction. The conclusion should also be supplemented by a fragment related to the problems and future outlook of acerola recycling from chemistry and technology point of view. This revision would be helpful to more fitting this manuscript to Recycling scope.

Comments on the Quality of English Language

English  is quite understandable in this manuscript.

Author Response

Reviewer 4:

Despite the manuscript title, there is very scarce information and consideration related to Environmental Preservation in the Manuscript. So, it should be supplemented by analysis of the modern literature related to acerola wastes recycling, e.g., https://doi.org/10.1016/j.wasman.2020.03.037 . Different ways of this recycling should be considered, for instance, thermochemical treatment affording biochar, and concentrate more on chemical and technological aspects of recycling.

We removed the environmental part from the title as it was not the text's focus. On the other hand, we added more sub-sections focusing on other uses of the acerola waste and added the reference suggested.

Line 618:

  1. Biofuels and energy applications

 Nogueira et al. (2019) explored the utility of Hydrothermal Carbonization (HTC) for optimizing the treatment and application of waste from acerola processing. Various factors, including temperature, time duration, biomass-to-water ratio, pH levels, and mixing speed, were evaluated for their impact on the efficiency of the HTC process and the properties of the resulting hydrochar. Optimal conditions were determined through a multi-criteria evaluation method. The hydrochar exhibited a higher concentration of oxygen-bearing functional groups than similar biomass types cited in existing literature. Further experiments assessed the hydrochar's effectiveness in absorbing methylene blue dye, revealing peak absorption and removal efficiency conditions. The data suggests that the hydrochar's absorption is heat-driven and spontaneous, indicating that acerola waste could be transformed via HTC into a valuable resource for absorbing environmental pollutants [112].

De Oliveira reported obtention of ethanol from the hydrolysis of baggase from acerola and further fermentation of the glucose obtained. They pre-treated them with diluted sulfuric acid and enzymatic hydrolysis with Trichoderma reesei, obtaining glucose ranging from 0.78–1.95 g/L. Saccharomyces cerevisiae was used for fermentation and ethanol production [113].

The anaerobic digestion of the waste can be a source of biogas and methane. Barros et al. reported the anaerobic co-digestion of acerola (Malphigia emarginata)  agro-industry effluent with domestic sewage at mesophilic and thermophilic conditions have been reported, achieving biodegradability above 60% at mesophilic conditions. CODS removals and their conversion into methane above 50% and 45% in mesophilic conditions, respectively [114].  Two methods to produce important energy gas sources, such as biological hydrogen by photo-fermentation, where the dominant biocatalyst is photosynthetic bacteria, and dark-fermentation, fermentative anaerobic bacteria produce hydrogen in the absence of light, have not been yet explored [115].

  1. Composites and nanomaterials

Leon, 2014 focused on synthesizing and characterizing a nanocomposite material comprising titanium dioxide and botanic and fruit extracts in nanoparticulate form (within a size range of 1-100 nm). This advanced material could be synthesized using an impregnation technique, wherein organic functional groups, inorganic moieties, and phytoextracts—such as could be the case with acerola extracts— could be adsorbed onto the titanium dioxide scaffold [116]. The result is an agent with pronounced antimicrobial properties, including disinfectant and antiseptic effects, that are efficacious against a broad spectrum of pathogens: bacteria, fungi, mycobacteria, spores, protozoa, and viruses. The product is a liquid suspension containing the solid nanomaterial in question.

The impregnation methodology is employed to achieve homogeneous dispersion of the functional groups and extract nanoparticles, while intramolecular interaction stability is maintained by temperature control during the synthesis process [116].

The antimicrobial activity profile of this conjugated nanomaterial is intrinsically linked to several parameters, including the particle size of the oxide scaffold, chemical functionalization, and surface dispersion of adsorbed extracts. Possible functional groups for incorporation include hydroxyl, carboxyl, amine, sulfate, and phosphate moieties. Furthermore, although the study focuses on titanium dioxide as the scaffold material, the technology could be extendable to other metallic oxides, such as silica, zirconia, zinc oxide, and alumina, without being exclusively restricted to these functional groups, extracts, or scaffolds. Therefore, this nanomaterial has applicability across diverse fields, from medicine to green technologies, offering an innovative and sustainable approach to pathogen management [116]. Reinaldo et al. reported the addition  of grape skins and acerola residues to the cassava thermoplastic starch, developed a bioactive packing  with  improving  antioxidant characteristics [117]

  1. Waste processing

  1. Silva et al. (2020) focused on utilizing ultrasound-assisted methods to extract these antioxidants from leftover acerola material. Various factors, like the concentration of ethanol in the hydroethanolic mix, duration of extraction, operational temperature, and the ratio of liquid to solid, were assessed for their impact on total phenolic and flavonoid content and antioxidant capabilities. Optimal conditions for extraction were determined using a multi-criteria optimization approach known as the desirability function. High-Performance Liquid Chromatography with Ultraviolet detection (HPLC-UV) was employed to pinpoint the primary antioxidant elements extracted under these conditions. The findings indicate that acerola byproducts offer promising avenues for enhanced utilization, particularly in the food and pharmaceutical sectors [118].

Nóbrega et al. (2015) investigated the impact of hot air drying on acerola waste's properties and nutritional content. The research explores various drying conditions, such as 60, 70, and 80°C temperature settings and air speeds of 4, 5, and 6 m/s, comparing them to untreated acerola waste [119]. The study quantifies the retention of key bioactive elements like phenolic compounds, carotenoids, anthocyanins, proanthocyanidins, and ascorbic acid after drying. The antioxidant properties were also measured. The findings indicate that the dried acerola waste retains significant levels of these important compounds, making it a viable source of health-promoting ingredients. This research offers a scientific foundation for the better use of acerola waste, especially given the growing issue of waste generation from agricultural production, notably in Brazil—one of the world's largest agricultural hubs. The dried acerola byproduct thus holds the potential for sustainable, health-focused applications in the food industry [119].

For another instance, there are studies to examine the impact of different agricultural practices on soil organic carbon (C) and nitrogen (N) levels in Chapada da Ibiapaba, Brazil [121]. It compares organic acerola fruit farming and pastureland with conventional crop rotation systems featuring carrot, beet, and corn. Soil samples were taken from various depths and analyzed for indicators, including total organic C and N, microbial C and N, and mineralizable C [121]. The findings show significant improvements in organic and pasture systems compared to conventional and native forest soil, particularly in microbial C and N stocks. For instance, the organic acerola system had a 585% increase in Nmic stock compared to native forest soil, while conventional farming saw reductions of up to 59% in Cmic stocks. These metrics suggest that organic and pasture management practices sustain and improve soil quality and carbon sequestration more effectively. Therefore, these sustainable approaches should be considered for better soil health in the Chapada da Ibiapaba region [120].

The authors should also to point out the position of this review among relevant literature in introduction. The conclusion should also be supplemented by a fragment related to the problems and future outlook of acerola recycling from chemistry and technology point of view. This revision would be helpful to more fitting this manuscript to Recycling scope.

We modified the summary to highlight that this review discusses the opportunities from a broader industry type perspective:

The study reviews the many uses for waste generated from acerola (Malpighia spp.) production, a tropical fruit renowned for its nutrient-rich content. Traditionally considered an environmental burden, this waste is now gaining attention for its sustainable applications in green technology.  This review outlines the extraction of valuable bioactive compounds like polyphenols, carotenoids, and pectin that can be extracted from the acerola fruit and acerola waste, and it also delves into its potential in material science, particularly in the creation of pharmaceutical formulations, nanomaterials, composites, biofuels, and energy applications. On the medical front, the paper highlights the promise that acerola waste holds as anti-inflammatory, anti-hyperglycemic activity, and anticancer therapies. By outlining challenges and opportunities, the review emphasizes the untapped potential of acerola waste as a resource for high-value products. These findings suggest a paradigm shift, turning what has been considered waste into a sustainable asset, thereby encouraging environmentally responsible practices within the fruit industry.

We modified the conclusions to:

Acerola (Malpighia spp.) is an economically significant tropical fruit, primarily due to its rich content of antioxidants, particularly ascorbic acid. This study emphasizes the fruit's diverse applicability, extending from the food industry to energy and biofuels, with emerging possibilities in natural medicine and cosmetics. Although Brazil is at the forefront of global acerola production, large-scale cultivation yields considerable waste, accounting for approximately 40% of the fruit's total volume.

Rather than merely posing an environmental challenge, these residues stand as an untapped resource for sustainable development. Our findings indicate that these by-products are abundant in bioactive compounds such as phenolics and other phytonutrients. Moreover, burgeoning research is unveiling the health benefits of acerola beyond its antioxidant capacity, including anti-inflammatory and antimicrobial properties. We did not identify the processing as a significant challenge because fruit and vegetable companies have created processes to reutilize this type of waste. The acerola waste is non-toxic and almost 100 reusables as a raw material for new products, as has been demonstrated by flour production; the primary challenge is to separate the bioactive compounds of interest without degrading them and/or to find synergies with other stabilizing compounds to create long lasting products . Improving the management of acerola waste due to its biodegradability and water content is a key factor that can catalyze the creation of new products and markets for these residues. Also, the consumer sensibilization to buying products from sources considered waste without losing efficacy.

It is critical to highlight that these residues could serve as valuable assets within green technology, with potential applications in bioactive compound extraction, biopolymer development, and biofuel generation. Therefore, the effective utilization of acerola waste addresses environmental challenges and introduces economic value to what has traditionally been considered waste.

Sustainable management systems focused on using acerola and its by-products could play a significant role in environmental conservation and soil quality improvement and contribute to carbon sequestration in specific regions. This study underscores the necessity for additional research to bridge the gaps in our understanding of acerola's multifaceted applications and its potential in sustainable waste management.

The present investigation is a valuable resource for future research, providing scientific justification for the more effective exploitation of this tropical fruit, aligning agricultural production with sustainability goals and added economic value.

Round 2

Reviewer 1 Report

Comments and Suggestions for Authors

The authors corrected the manuscript in accordance with the suggestions of the reviewer.

Author Response

Thanks a lot

Reviewer 2 Report

Comments and Suggestions for Authors

The writers have addressed all of my issues, and I have only found a few typographical errors in Fig1.   

Comments on the Quality of English Language

-

Author Response

Thanks a lot.  I attached the corrected figure. 

Reviewer 3 Report

Comments and Suggestions for Authors

The authors have enhanced the manuscript by incorporating supplementary data pertaining to the economic and practical facets associated with the utilization of acerola. Furthermore, they have introduced novel perspectives regarding the application of acerola, such as its utilization in biofuels, energy applications, cosmetics, among others. Following these substantial modifications, I hereby approve the manuscript for publication.

Author Response

(The authors gave the same response as above.)

Reviewer 4 Report

Comments and Suggestions for Authors

The authors revised their manuscript according all reviewer's comments.

It will be publish in present form.

Comments on the Quality of English Language

English in this paper is quite understandable.

Author Response

Thanks a lot